

# 1  Assessment of the aerosol optical depths measured by satellite-
# 2  based passive remote sensors in the Alberta oil sands region

Christopher E. Sioris[1], Chris A. McLinden[1], Mark W. Shephard[1], Vitali E. Fioletov[1], and Ihab
Abboud[1]
[1] {Environment and Climate Change Canada (ECCC), Toronto, ON, Canada}
*Correspondence to*: Christopher E. Sioris (christopher.sioris@canada.ca)
**Abstract.** Several satellite aerosol optical depth (AOD) products are assessed in terms of their data quality in the
Alberta oil sands region. The instruments consist of MODIS (Moderate resolution Imaging Spectroradiometer),
POLDER (Polarization and Directionality of Earth Reflectances), MISR (Multi-angle Imaging SpectroRadiometer),
and AATSR (Advanced Along-Track Scanning Radiometer). The AOD data products are examined in terms of
multiplicative and additive biases determined using local AERONET (AEROCAN) stations. Correlation with
ground-based data is used to assess whether the satellite-based AODs capture day-to-day, month-to-month, and
spatial variability. The ability of the satellite AOD products to capture interannual variability is assessed at Albian
Mine and Shell Muskeg River, two neighbouring sites in the northern mining region where a statistically significant
positive trend (2002-2015) in $PM_{2.5}$ mass density exists. An increasing trend of similar amplitude is observed in this
northern mining region using some of the satellite AOD products.
**1  Introduction**
Fine-mode aerosols can be harmful to the respiratory system in large doses and are thus a critically important
constituent with regard to air quality. For this reason, particulate matter with median aerodynamic diameter less than
2.5 μm ($PM_{2.5}$) is one of the atmospheric observables used to calculate the Air Quality Health Index (AQHI) in
Canada (Stieb et al., 2008). Similar indices are used in other countries (Kelly et al., 2012). Tropospheric aerosols are
also a major source of uncertainty in estimating the radiative forcing of climate (Myhre et al., 2013). Many satellite-
based instruments can provide information about atmospheric aerosols in the form of aerosol optical depth (AOD), a
measure of the vertically integrated extinction of the solar beam by aerosols.  Measurements of AOD tend to be
proportional to particulate matter mass density measured at the surface when the boundary layer aerosol
concentrations are elevated (e.g. Tian and Chen, 2010).
The Alberta oil sands region (AOSR) has been under rapid industrial development during the past decade (Foote,
2012). Satellite measurements already indicate a significant increasing trend in nitrogen dioxide between 2005 and
2014 (McLinden et al., 2012; McLinden et al., 2016). Additionally, the AOSR is being deforested as part of
expanding surface mining operations. This inevitably increases levels of dust, which partly arises from



transportation by trucks. Dust is one of many aerosol types of relevance in the AOSR. Other main aerosol types
include organic aerosols, both natural and anthropogenic (Liggio et al., 2016), as well as ammonium sulfate.
Passive remote sensing of aerosol over land is challenging because, for a cloud-free scene, most of the nadir
radiance is coming from direct reflection off the surface at visible wavelengths, not from aerosol scattering. This is
particularly true for the AOSR, which consists of an irregularly-shaped industrial area to the south comprised of
non-vegetated (cleared) mining locations and a second area to the north where mostly surface mining is occurring,
as both areas have high surface albedo in the visible. Within the AOSR, the land type changes on spatial scales
smaller than the typical $10 \times 10$ km AOD footprint of a satellite-based instrument. Considering the area surrounding
the AOSR, specifically the rectangular area between 55.0 and 58.5°N and 114.0 to 108.5°W, the land is covered by
evergreen needleleaf forest (70%) and some deciduous broadleaf forest (23%), which is typical of the boreal forest
in the northern portions of the Alberta and Saskatchewan.
**2   Method**
In order to study the spatiotemporal distribution of AOD in the AOSR, data from several satellite-based instruments
are used. Satellite-based aerosol sensors are chosen based on a number of factors. One of the goals of the study is to
examine long-term AOD trends, so preference is given to instruments with longer data records. Instruments that
view a scene with multiple viewing angles were selected as the multi-angle capability is useful for disentangling the
contributions to the scene reflectance by the surface and by the overlying aerosols (e.g. Bevan et al., 2012). Such
instruments include Multi-angle Imaging SpectroRadiometer (MISR) (Diner et al., 1989), the Polarization and
Directionality of Earth Reflectances (POLDER) series (Deschamps et al, 1994) including POLDER/PARASOL
(Polarization & Anisotropy of Reflectance for Atmospheric Sciences coupled with Observations from a Lidar), and
the Along-Track Scanning Radiometer (ATSR) series (see Table 1 for the spatial resolution, temporal coverage and
wavelength at which AOD is reported for each of the satellites). In addition, MODIS (Moderate resolution Imaging
Spectroradiometer) is chosen partly because it has a long wavelength channel (2.1 μm) that allows the surface
reflectance to be accurately determined over vegetation without contamination from fine-mode aerosols (e.g.
particles with radii of <0.2 μm) by virtue of the correlation between visible and 2.1 μm surface reflectance for
vegetation (e.g. Kaufman et al., 2002; Li et al., 2005). MODIS/Aqua collection 6 data are used (see Appendix for
providers and version numbers of other satellite data products). For MODIS, there are two AOD retrieval algorithms
yielding the Dark Target (DT) (Levy et al., 2013) and the Deep Blue (DB) (Hsu et al., 2013) products. Specifically,
the Corrected_Optical_Depth_Land (470 nm) and the Deep_Blue_Aerosol_Optical_Depth_550_Land datasets were
used. The Dark Target algorithm exploits the fact that, for dark surfaces, aerosols tend to brighten the scene. For
highly reflective surfaces such as snow in the visible spectral region, AOD cannot be retrieved using either the DT
or DB approach.  The MODIS Aqua DT product is also processed at 3 km spatial resolution in addition to the
standard 10 km resolution available for both MODIS products (Levy et al., 2013). Each MODIS AOD measurement
is assigned a confidence value.  Confidence values of 1 and 0 indicate marginal and no confidence, respectively,
while values of 2 and 3 represent good and ideal confidence (Levy et al., 2013). For MODIS/Aqua collection 6, data
with confidence≥1 are retained for validation. The theoretical basis of the MISR aerosol retrieval algorithm is given



by Diner et al. (2008). The aerosol retrieval for AATSR is described by Bevan et al. (2012) and references therein.
Deuzé et al. (2001) detail the approach used to retrieve aerosol information from POLDER observations over land.
MODIS Terra is not considered since it is highly similar to MODIS Aqua but, for collection 6, the former is less
reliable for trend studies in spite of improvements relative to collection 5 (Levy et al., 2015). The MODIS-based
Multiangle Implementation of Atmospheric Correction (MAIAC) (Lyapustin et al., 2011) product is not currently
available in the AOSR (van Donkelaar et al., 2016). VIIRS (Visible Infrared Imaging Radiometer Suite) (Hillger et
al., 2013) is not considered in this study because of its shorter data record relative to the MODIS sensors. Active
remote sensing instruments are not considered because of the long revisit time and poor spatial coverage of the
relatively small AOSR.
For validation of satellite-based AOD data, AERONET (Holben et al., 1998) is the ideal choice since the same
quantity is measured by this ground-based network of direct-sun multiband photometers and the ~3 minute typical
sampling interval generally ensures a good temporal coincidence during clear sky conditions. Quality-controlled
AERONET data (Level 2, version 2) are used (http://aeronet.gsfc.nasa.gov). CIMEL (French manufacturer) CE318
sensors used by AERONET measure at several wavelength, some of them (e.g. 500 and 870 nm) are close to the
wavelengths at which the selected satellite instruments report AOD (e.g. 470, ~550, and 865 nm). There are two
AERONET sites in the oil sands region: Fort McMurray (56.752°N, 111.476°W) and Fort McKay (57.184°N,
111.64°W). Measurements at Fort McMurray started in 2005. The Fort McKay site has only been in operation since
August 2013 meaning that there is no temporal overlap with Advanced ATSR (AATSR) and only seven
coincidences with POLDER/PARASOL using coincidence criteria of ±12 minutes and 10 km. The spatial
coincidence criterion corresponds to the smallest AOD footprints of the selected data sets (Table 1). A larger spatial
coincidence criterion is not considered since, as shown below, strong spatial gradients in AOD exist in this aerosol
source region. Furthermore, as mentioned in Sect. 1, the surface type also changes on such spatial scales. The
temporal coincidence criterion was set to limit the number of independent AERONET measurements used in the
statistical analysis. There can be multiple AERONET observations that are temporally coincident with a satellite
observation and there can be up to four spatial coincident satellite AODs during a satellite overpass of an
AERONET site. All of these coincidences are treated as independent data points in the validation and correlation
analyses. In order to properly validate satellite AOD bias, AERONET 500 nm AODs are interpolated to the satellite
AOD wavelengths (see Table 1) using the coincident AERONET Ångström exponent derived from 440 and 675 nm
measurements, except for POLDER/PARASOL, for which no scaling of the AERONET was applied.
The ability of each satellite-based sensor to capture the AOD seasonality in snow-free months is determined at Fort
McMurray using the correlation of monthly averaged AODs (using all overlapping years) with AERONET. A
minimum of 20 coincident data points per calendar month must be available for that month to be included in the
correlation.
In order to assess the ability of the satellite data to capture the spatial variability in this region, spatial correlation is
determined for hourly in-situ surface-level PM2.5 from the 10 NAPS (National Air Pollution Surveillance) stations
(Table 2) and satellite AODs averaged over all coincidences within their temporal overlap period. NAPS stations





continuously monitor $PM_{2.5}$ mass density using tapered element oscillating microbalances (TEOMs). The NAPS
network is reviewed by Demerjian (2000), although recently there has been a gradual shift in technology since 2011
to a SHARP (Synchronized Hybrid Ambient Real-time Particulate) monitoring system, which is a hybrid of a
nephelometer and a beta attenuation monitor (Hsu et al., 2016). The same 10 km spatial coincidence criterion is used
but temporal coincidence limit is extended to ±1 hour to match the temporal resolution of the selected NAPS
datasets.
Similar to the spatial and seasonal variability, the ability of the satellite instruments to capture interannual variability
can be assessed by correlating yearly satellite-based AODs averaged over all coincidences with NAPS $PM_{2.5}$
measurements over the overlap period. 20 coincidences in a calendar year are required for the year to be included in
the correlation calculation. As an example, for MISR, 14 sufficiently sampled years (2002-2015) are used in the
correlation with NAPS data at Millennium mine.
For temporal trends in AOD, a simple linear regression is performed on annual averages and medians. Similarly, for
$PM_{2.5}$, the annual average of daily average values are used since the $PM_{2.5}$ auto-correlation timescale is on the order
of 6.5 hours, based on analysis of Albian mine $PM_{2.5}$ data from 2002. The extra step of daily averaging prior to
annual averaging yields more conservative annual standard error (s. e.) estimates. Partial years at the start and the
end of a data record are removed. Trend periods are given below for each sensor. The area over which the satellite-
based AOD trend maps are calculated is 0.1°×0.1° by default. This default setting is used to determine the AOD
trend for both MODIS/Aqua 10 km products (2003-2015). The trend domain considered in this work spans from 56-
58°N and 111-112°W. For sensors with poorer spatial coverage (MISR, AATSR, POLDER/PARASOL), the spatial
binning is expanded in latitudinal and longitudinal increments of 0.1° until there are ≥20 observations in each
calendar year within at least one grid cell in the domain. The trend maps are ultimately generated at 0.3°×0.3° for
AATSR (2003-2011) and MISR (2000-2015) whereas a 0.4°×0.4° area is required for POLDER/PARASOL (2005-
2013). Outlying individual data points (>4 standard deviations above the climatological average in the domain) are
recursively filtered mainly to reduce the influence of forest fires on trends. The same filtering is applied to the $PM_{2.5}$
datasets. Interannual consistency in the month-to-month sampling is checked for any location with a positive
satellite AOD trend significant at the 95% confidence interval by calculating the average day-of-the-year for each
calendar year. Such temporal sampling anomalies occur for MISR AOD data at some locations if a 0.1°×0.1° grid
were used, for example. The Albian mine (2001-2008) and Shell Muskeg River (2009-2015) forest-fire-filtered
$PM_{2.5}$ datasets were merged for trend analysis since the sensor was relocated from the former to the latter site in
January 2009 and these sites are separated by <5 km.

**3   Results**
First, the general spatial distribution of AOD is illustrated for some of the aforementioned data sets. In Fig. 1, the
climatological average POLDER AOD on a 0.1° × 0.1° grid is shown. This is the default grid used for
climatological maps of all satellite AOD datasets. The POLDER sample size per grid cell is 90 to 170 in the AOSR
over the discontinuous period from 1996 to 2013 (see Table 1). There is a clear hotspot in 865 nm AOD in the





AOSR region, roughly double the surrounding background values. Note that for POLDER and MISR, there are
expected voids in their spatial coverage (Fig. 1) due to the spatial sampling of these instruments, whereas MODIS
and AATSR footprints can be centered on any geolocation within the AOSR.
The AOD hotspot in the AOSR seen by POLDER is less obvious with MISR (Fig. 1). The ability to capture spatial
variability with MISR is generally much worse than the other instruments based on spatial correlations of average
satellite-based AOD versus average NAPS $PM_{2.5}$ mass density over the ~10 available sites (Table 3). Table 4
provides the number of coincidences for each satellite with the both Fort McMurray and Fort McKay AERONET
observations to provide a relative sense of the sample sizes.
The climatological AOD maps for the MODIS/Aqua collection 6 DT and DB products (2002-2014) are also shown
in Fig. 1 however there is a major issue with the confidence as shown in Fig. 2. Near the Syncrude facility at
Mildred Lake (57.05°N, 111.6°W), the confidence approaches 0 in both MODIS products in the two adjacent
0.1°×0.1° cells (Fig. 2). In the western cell, the inadequate confidence in MODIS Aqua collection 6 DT data is due
to failure of the AOD retrieval algorithm due to the 2.1 µm reflectance exceeding the allowed upper limit of 0.35.
This is a fundamental weakness of the Dark Target retrieval strategy (see sect. 2). In the adjacent eastern cell, the
low confidence stems from the low number of $0.5 \times 0.5$ km$^2$ pixels (see Table 1) used in the AOD retrieval. The
number of pixels used in the AOD retrieval is reduced by high 2.1 µm reflectance (>0.35), but also by cloud
masking and an independent test for optically thicker cirrus, diagnosed using the 1.38 µm channel (Levy et al., 2013;
Hubanks, 2015). The high reflectance in the near-infrared affecting the western cell and possibly the eastern cell is
typical of desert or sandy loam. The higher spatial resolution of the MODIS-Aqua 3 km DT data clarifies the
importance of this issue: key areas in the AOSR are simply not monitored with confidence by the current
MODIS/Aqua DT product. For example, there are 0.01° × 0.01° areas with no AOD measurements of the highest
confidence in 12 years, whereas surrounding, equal areas have tens of observations. The lack of confidence is not
unique to the AOSR. Low confidence is also observed in urban areas within the province (e.g. Calgary, not shown).
The low confidence in the MODIS DB product is due to the spatial heterogeneity of the surface between vegetated
and non-vegetated area, which leads to pixels falsely identified as cloudy (N. Christina Hsu, NASA, priv.
communication). Li et al. (2009) identified the need for improved AOD measurements using the DB algorithm over
transitional land covers.
A similar issue exists for AATSR (Fig. 3) and ATSR-2 (not shown), which both have an exceedingly small number
of successful retrievals in a 0.1° × 0.1° area containing the Mildred Lake Syncrude facility (e.g. N<10) during their
respective missions (Table 1). Similarly to MODIS, this is probably caused by falsely identifying bright patches in
otherwise vegetated scenes as clouds (P. North, Swansea University, priv. communication). Cloud fraction for
successful AOD retrievals tends to be as high as 0.18 within the oil sands region, including the northern mining
region, yet drops to 0.02 in the surrounding region (Fig. 3). Note that cloudy 1 × 1 km$^2$ pixels are not used during the
AATSR AOD retrieval.  The spatial correlation coefficient between sample size and cloud fraction as illustrated in
Fig. 3 is -0.73, indicating that the spatial variation in AATSR sample size is mostly related to cloud flagging.





Neither POLDER nor MISR show a sampling void in the AOSR. Table 1 shows that these two sensor types have
coarser AOD spatial resolution by a factor of 3-4 than MODIS, ATSR-2, and AATSR. Note that some of the $PM_{2.5}$
sites are located in the periphery of the industrial and mining areas and thus spatial coincidences exist for MODIS
and AATSR in spite of the aforementioned issues, given the 10 km coincidence criterion.
In terms of the validation using AERONET data (Table 4), MISR has a large multiplicative bias (i.e. small slope),
which is consistent between both sites in the AOSR. Excluding Fort McMurray coincidences for which the
AERONET AODs interpolated to 558 nm are >0.4, the slope improves to 0.74 and is of a similar value to the slope
found in previous studies for inland (Liu et al., 2004), dusty (Kahn et al., 2005), and urban environments (Jiang et
al., 2007). MODIS DB tends to yield more data than the DT product, but the correlation is lower with AERONET
on individual coincidences and in terms of the seasonal variation. At both AERONET sites, the MODIS products
behave oppositely in terms of multiplicative and additive biases (discussed in Sect. 4). AATSR and
POLDER/PARASOL show no major deficiencies, with the latter exhibiting the closest slope value to unity of all of
the satellite sensors at Fort McMurray.
**3.1    Trends**
Before considering trends in the AOSR, it is useful to look at whether the different satellite data products capture the
AOD interannual variability at Fort McMurray, where a sufficiently long record (2005-2015) of 500 nm AOD
exists. All of the products capture the interannual variability of the annual mean AOD observed by AERONET at
Fort McMurray (Table 5). Correlation coefficients for forest-fire-filtered annual means tend to be only slightly
lower.
In general, very few of the 200 grid cells in the trend domain (56-58°N, 111-112°W) indicate a statistically
significant (2 s. e.) positive trend that is consistent from one satellite to the next. In fact, there are no points in the
domain for which MODIS/Aqua DT (2003-2013), AATSR, or ATSR-2 (1996-2002, 0.3°×0.3°) show a significant
positive trend in AOD. Similarly, POLDER/PARASOL only shows a significant positive trend in three adjacent grid
points at 57.3°N between 111.3 and 111.5°W (see Fig. 4) and MISR also finds a significant positive trend at only
two locations in the domain. Finally, MODIS/Aqua DB has two points with the largest and most significant positive
AOD trend in the region of the Muskeg River mine at 57.25°N, 111.25°W (Fig. 4). In fact, two satellite data
products, namely POLDER/PARASOL and MODIS/Aqua DB, exhibit a significant positive trend in this mining
area. Although not statistically different from zero, the AOD trend in both AATSR and MISR data is positive in the
area of the positive POLDER/PARASOL trend (Fig. 4), whereas MODIS DT tends to show an insignificant
negative trend.
Changes to the surface may be at the root of the increasing AOD trend in this area, either since clearing of
vegetation could lead to higher concentrations of dust, or by biasing the AOD retrieval. Trends in surface albedo
were determined from the combined MODIS Terra/Aqua MCD43C3 albedo data product at four wavelengths
relevant to the MODIS or POLDER AOD retrievals: 470, 645, 860, and 2130 nm (see Appendix A). For all four
wavelengths, neither the largest nor the most significant trends in surface reflectivity occur at 57.25°N, 111.25°W





(not shown), where the largest and most significant MODIS DB AOD trend occurs and also within the larger area of
the spatially coherent POLDER/PARASOL AOD trend.
In order to quantitatively compare trends in AOD and $PM_{2.5}$, the ratio of the average AOD to average $PM_{2.5}$ mass
density over all coincidences between each satellite instrument and a given NAPS site is used to convert the AOD
trends from the satellite instruments to $PM_{2.5}$ trends. This implicitly assumes that the ratio of $PM_{2.5}$ to AOD is
constant over time. This ratio is determined for the merged Albian mine / Shell Muskeg River dataset. Since aerosol
optical depth histograms indicate a skewed distribution, it is also useful to verify trends using annual medians. For
that purpose, the ratio of median AOD to median $PM_{2.5}$ is used instead. This approach is particularly important for
POLDER/PARASOL because of the very low 865 nm AODs (Fig. 1) and the negative offset (Table 4) that do not
allow a relative trend to be meaningful.
A significant positive trend of 0.24±0.06 (±1 standard error) (Figs. 5-6) and 0.24±0.07 μg/m$^3$/year is detected in the
Albian mine/ Shell Muskeg River merged annual average and median $PM_{2.5}$ mass densities (2002-2015),
respectively. Limiting the merged $PM_{2.5}$ dataset to the warm season (April-October) to mimic the temporal coverage
of the satellite data (Table 4), the trend (0.25±0.07 μg/m$^3$/year) does not change significantly from the trend using
year-round data (Fig. 6). A consistent trend of 0.21±0.09 μg/m$^3$/year is found in annually-averaged $PM_{2.5}$ at Albian
mine (2002-2008) alone, and the trend there during the warm season is also statistically significant and not different
(0.24±0.06 μg/m$^3$/year). Furthermore, there is no indication of a discontinuity between 2008 and 2009 when the
monitoring site was relocated. The trend in $PM_{2.5}$ at the surface is in quantitative agreement with the $PM_{2.5}$ trends
derived from MODIS/Aqua Deep Blue and POLDER/PARASOL annually averaged AOD data over similar, yet
shorter periods. For both MODIS/Aqua Deep Blue and POLDER/PARASOL, trends using annual medians agree
with trends determined using annual averages within their respective standard errors (1 s. e.). The low bias of
POLDER/PARASOL AOD near these two Shell mines is expected from the validation with AERONET at Fort
McMurray (Table 4) and previous work on larger spatial scales (Deuzé et al., 2001).
Contrary to the localized, significant AOD trend in satellite data records in the eastern portion of the Muskeg River
region, a statistically significant trend is found at two other ground-based stations within the AOSR for the period
2002-2014, namely Syncrude UE1 and Millennium mine (Fig. 6). The largest trend occurs at Millennium mine, the
closest NAPS station to the southeast of the Shell Muskeg River region (see Table 2 and Fig. 4 for location). The
trend is insignificant using either annual means or median $PM_{2.5}$ data at CNRL Horizon and Anzac where data
records are shorter, while the trend at Wapasu (2013-2015) was not evaluated. The $PM_{2.5}$ trends at the remaining
sites in the AOSR, namely two sites at Fort McMurray and one at Fort McKay are discussed below. Note that
POLDER/PARASOL does not measure at Syncrude UE1 (see Table 3) and there is insufficient sampling at
Millennium Mine over an area of 0.4°×0.4° in each of the years (2005-2013) for trend analysis. For
POLDER/PARASOL, the trend, while mostly insignificant in the AOSR, is always positive. For AATSR, the AOSR
has regions of statistically insignificant negative and positive trends. For MISR, the trend is positive in 56% of the
trend domain and even more so (83%) in the northern half of the domain (57-58°N). For MODIS DB and DT, some
of the AOSR is not sufficiently sampled with high confidence (see Sect. 2), but where confidence is ≥1, the trend





tends to be negative in 69% and 77% of this area, respectively. Bari and Kindzierski (2016) found no indications of
a positive trend in $PM_{2.5}$ at Fort McKay and the Fort McMurray Athabasca Valley site, using a longer period (1998-
2014), although, as shown in Fig. 2 of Bari and Kindzierski (2016) for Fort McKay, there is an abrupt decrease in
$PM_{2.5}$ mass densities that occurs between 2001 and 2002 that has a profound effect on the trend and its uncertainty.
This discontinuity is observed at all sites in the AOSR that extend back to 2001. An earlier study by the same
authors (2015) also indicated no trend between 1998-2012 at the same sites and at the Fort McMurray Patricia
MacInnes site as well. Li et al. (2016) find a small positive trend in AOD over Athabasca (56-58°N, 110-113°W)
using MODIS/Aqua DB data (2004-2015), insignificant at the 2 s. e. level.
**4    Discussion and conclusions**
In this section, the advantages and limitations of the various data products are summarized. As shown in Table 4, all
of the satellite sensors capture the temporal variability in AOD over Fort McMurray, based on correlations with
AERONET, in spite of the low AODs there (e.g. Fig. 1). This temporal variability is largely driven by day-to-day
variability as forest fires lead to episodes with large AODs (>3) in summer months that strongly influence the
calculated correlation.
The two MODIS AOD data products (Deep Blue and Dark Target) have low confidence in the AOSR due to issues
relating to elevated surface reflectivity in the vicinity of the Mildred Lake Syncrude facility. However, the MODIS
dark target product is the best at capturing temporal variability in terms of the correlations with AERONET AOD at
Fort McMurray and in terms of capturing the month-to-month variability. This is likely due to MODIS's
combination of spatial resolution (Table 1) and higher signal-to-noise ratio (SNR): its radiances have SNR > 1000
(Xiong et al., 2003) whereas the other instruments have SNR of 1000 or less (Deschamps et al., 1994; European
Space Agency, 2007; Diner et al., 1989). MODIS DT clearly has a slope slightly greater than unity over the AOSR,
in contrast to MODIS DB (Table 4). Focussing on Fort McMurray, where there is a longer AERONET data record
than at Fort McKay, the MODIS DT slope changes insignificantly when coincident AERONET AOD is limited to
<0.7. The same pattern of consistently high and low slope values for the MODIS Aqua DT and DB (collection 6)
products, respectively, was found over two sites in Pakistan, namely Lahore and Karachi, by Bilal et al. (2016) and
during non-fire summertime periods over semi-arid Nevada and California as shown in Table 4 of the work of
Loría-Salazar et al. (2016). A high slope may be related to the use of the 2.1 µm channel to determine the reflectivity
in the visible over non-vegetated surfaces as suggested by Bilal et al. (2016). High-biased AODs result because the
surface reflectance in the visible assumed by the retrieval algorithm is less than the actual value as the relationship
between the visible and 2.1 µm was developed for vegetated land for which a stronger spectral variation exists than
for barren land. Li et al. (2005) have shown that the spectral reflectance relationship is much different even for dry
vegetation than green vegetation. Note that high day-to-day variability can be captured in spite of biases in assumed
surface reflectance since the latter changes slowly with time over the warm season, when successful measurements
occur more frequently. A MODIS algorithm designed to function over inhomogeneous surfaces such as the AOSR
region, and which would also likely be applicable to urban areas, is being investigated to exploit the many benefits
of MODIS radiance data. One such benefit is the twice-daily revisit over the AOSR that the current multi-angle





sensors, namely MISR and SLSTR (Sea and Land Surface Temperature Radiometer) (Coppo et al., 2010), cannot
offer. SLSTR, onboard the recently launched Sentinel-3a satellite, is the next generation in the ATSR series.
MISR clearly captures the short-term and month-to-month AOD variability at Fort McMurray based on correlations
at the individual coincidence level and the monthly time scale (Table 4), but struggles to capture the local spatial
variability including the AOD hotspot in the AOSR as discussed in Sect. 3. The MISR low bias may be related to the
need for darker spherical particles (Kahn et al., 2005) given that forest fire smoke plays a significant role throughout
the western Canada in the warm season (O'Neill et al., 2002). Spherical particles with lower single scattering albedo
(SSA) may also be required to properly represent local anthropogenic pollution (Kahn et al., 2005) in the AOSR.
The 3×3 superpixel averaging that is used when the MISR retrieval fails for the central superpixel could also
contribute to a low bias (Jiang et al., 2007), particularly at Fort McKay as background AODs to the west could be
lowering the average.
AATSR has a major spatial sampling issue in the heart of the AOSR, but also captures month-to-month variability
from late spring to early autumn (Table 4) as well as short-term (Table 4) and spatial variability (Table 3). Based on
a previous analysis (Che et al., 2016), the AATSR AOD underestimation of the Swansea University product (also
used here) is larger over barren surfaces or sparse vegetation. Such land cover types are present in the AOSR. The
slight bias (Table 4) is not strongly AOD-dependent as removing coincidences with AERONET 500 nm AOD of
>0.35 does not significantly change the slope of the regression equation (Table 4).
POLDER has a known negative offset in AOD (Deuzé et al., 2001), confirmed using coincident Fort McMurray
AERONET AOD data. However, POLDER/PARASOL is the most accurate satellite-based aerosol sensor at Fort
McMurray during periods of the higher AODs (e.g. ≥0.31, Table 4), when its negative offset becomes rather trivial.
Overall, the POLDER AOD product is without a major weakness relative to the other instruments, although it is
provided at a relatively coarse spatial resolution (Table 1) and the fixed spatial sampling pattern of this sensor
inhibits the application of spatial oversampling techniques. The use of polarized radiances reduces the sensitivity of
the retrieved AOD to surface reflectance (e.g. Deuzé et al., 2001). The trend in POLDER/PARASOL AOD at the
Shell mines (Albian and Shell Muskeg River) is probably not driven by a trend in surface reflectance since
agreement with AERONET tends to be independent of surface type (e.g. Chen et al., 2015). A future sensor of
POLDER heritage, namely the Multi-viewing, Multi-channel, Multi-polarisation Imager (3MI), offers higher spatial
resolution, the availability of longer wavelength channels, and the potential for accurate monitoring of the local
aerosol loading in the decade to come.
While AODs in the AOSR are relatively small according to POLDER/PARASOL (Fig. 1), the significantly positive
trend in AOD from this satellite sensor and the similar trend in observed surface-level $PM_{2.5}$ in the region of the
Muskeg River mine points to the need to continue monitoring of this region with a combination of surface and
satellite-based aerosol observations.





*Acknowledgements.* Helpful discussions with Shailesh Kharol (ECCC) on the size range of dust particles are
gratefully acknowledged. The European Space Agency Climate Change Initiative program is acknowledged. Peter
North (Swansea University) is thanked for comments on the manuscript.

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

**Appendix A: Data product notes**
MODIS data is obtained from ftp://ladsweb.nascom.nasa.gov/allData/. AATSR and ATSR-2 version 4.1 data are
from Swansea University and can be obtained from the Aerosol CCI website (http://www.esa-aerosol-cci.org/)
following registration. The current file version (F12) is used for MISR
(ftp://l5eil01.larc.nasa.gov/MISR/MIL2ASAE.002). The selected MISR AOD product is named the "regional best
estimate of spectral optical depth". POLDER data was obtained from CNES (http://polder.cnes.fr), but data can
currently be obtained from http://www.icare.univ-lille1.fr/ following registration. A POLDER AOD datum is
filtered if any of the following statements are true (see F.-M. Bréon, 2011):
1)   The central pixel is snow-covered.
2)   One of the cloud tests is not applied.
3)   None of the 9 radiance pixels which form the AOD superpixel has clear sky.
4)   Sufficient data couples do not exist. The couples are:

28        a)   865 nm & 910 nm,

29        b)   Q443 & U443,

30        c)   Q670 & U670,

31        d)   Q865 & U865,

32          where Q and U are the derived Stokes elements and the number is the wavelength (in nm) of the

33          channel.

34      5)   Ozone absorption is not corrected (using TOMS or ECMWF).



6) Stratospheric aerosol correction is uncertain or imprecise (i.e. stratospheric AOD larger than a certain threshold).

7) Minimum scattering angle is larger than a threshold or maximum scattering angle is smaller than a threshold.

8) Aerosol optical thickness is larger than a threshold such that surface reflectance cannot be estimated adequately.

9) A large difference between measured and modeled reflectance exists for 443 nm.

10) Differences are too large between measured and modeled reflectance (risk of glitter).

11) Meteorological data indicate the presence of snow at ground level.

12) The quality index is 0.00 for viewing geometry conditions

13) The quality index is 0.00 for polarized reflectance fit.



| Satellite | Time period | Wavelength (nm) | Spatial resolution of AOD superpixel (km$^2$) | Spatial resolution of radiances (km$^2$) |
|---|---|---|---|---|
| MISR | 2000-2015 | 558 | 17.6 × 17.6 | 1.1 × 1.1 |
| MODIS: Terra | 2000-2015 | 470, 550, 660 | 10 × 10 (also 3 × 3) | 0.5 × 0.5 |
|       Aqua | 2002-2015 | | | |
| POLDER:   1 | 1996-1997 | 865 | 18 × 21 | 6 × 7 |
|         2 | 2003 | | | |
| (PARASOL) 3 | 2005-2013 | | | |
| ATSR: ATSR-2 | 1995-2003 | 550 | 10 × 10 | 1 × 1 |
|    AATSR | 2002-2012 | | | |

2  **Table 1**. Spatial resolution of AOD data products from selected satellite instruments. The third column contains the

3  wavelength at which aerosol optical depth is reported in each satellite data product. MISR and both MODIS

4  instruments are currently operating.



| Station name | lat(°N) | lon(°W) | Time span |
|---|---|---|---|
| Anzac | 56.4493 | -111.0372 | 2006-2015 |
| Fort McMurray Athabasca Valley | 56.7328 | -111.39 | 1997-2015 |
| Fort McMurray Patricia McInnes | 56.7522 | -111.476 | 1999-2015 |
| Millennium mine | 56.97 | -111.4 | 2001-2015 |
| Syncrude Upgrader Expansion 1 | 57.1492 | -111.642 | 2002-2015 |
| Fort McKay | 57.1894 | -111.641 | 1997-2015 |
| Wapasu | 57.2383 | -110.9028 | 2013-2015 |
| Shell Muskeg River | 57.2491 | -111.508567 | 2009-2015 |
| Albian mine | 57.2808 | -111.526 | 2001-2009 |
| Canadian Natural Resources Ltd. Horizon | 57.3037 | -111.739617 | 2008-2015 |

2    **Table 2**. Selected NAPS PM$_{2.5}$ sites and time span of available data (inclusive)



| AOD product | R | N |
|---|---|---|
| POLDER/PARASOL 865 nm | 0.83 | 8 |
| AATSR 550 nm | 0.77 | 9 |
| MISR 558 nm | -0.41 | 10 |
| MODIS/Aqua DT 470 nm | 0.49 | 10 |
| MODIS/Aqua DB 550 nm | 0.81 | 10 |

**Table 3**. Spatial correlation between $PM_{2.5}$ mass density and AOD using means of coincident data over the entire
overlapping period at 10 sites in the AOSR. Wapasu has insufficient or no temporal overlap with
POLDER/PARASOL and AATSR. Syncrude UE1 is not spatially coincident with any of the POLDER locations
given the 10 km criterion (see Sect. 2).



|  | R | slope | offset | seasonal r | month range | N |
|---|---|---|---|---|---|---|
|  | 0.81 | 0.89 | 0.0304 | 0.84 | 4-10 | 5508 |
| Aqua DB v6 | 0.94 | 1.00 | 0.0171 | 0.84 | 4-10 | 626 |
|  | 0.956 | 1.11 | -0.0013 | 0.99 | 4-10 | 4748 |
| Aqua DT v6 | 0.972 | 1.08 | -0.0177 | 0.959 | 5-9 | 408 |
|  | 0.92 | 1.09 | -0.03 | 0.89 | 5-10 | 414 |
| PARASOL | - | - | - | - | - | - |
|  | 0.91 | 0.88 | 0.0265 | 0.96 | 5-10 | 560 |
| AATSR | - | - | - | - | - | - |
|  | 0.89 | 0.63 | 0.0293 | 0.88 | 3-9 | 337 |
| MISR | 0.93 | 0.64 | 0.0364 | - | - | 87 |

**Table 4**. Statistical comparison of coincident AODs observed by satellite-based sensors and AERONET CIMEL sun
photometer. For each satellite AOD product, the upper row is for Fort McMurray and the lower row is for Fort
McKay. The CIMEL 500 nm AOD is used for comparison with all satellite sensors except POLDER/PARASOL, for
which the CIMEL 870 nm AOD is more appropriate (see Table 1). The simple linear regression equation used to
obtain the slope and offset assumes AERONET AOD and satellite-based AOD are the independent and dependent
variables, respectively. The number of MISR-Fort McKay coincidences is insufficient to assess the month-to-month
variability.





|  | Including +4σ outliers | Excluding +4σ outliers |
|---|---|---|
| POLDER/PARASOL | 0.995 | 0.81 |
| MISR | 0.91 | 0.94 |
| AATSR | 0.98 | 0.92 |
| MODIS DT | 0.97 | 0.95 |
| MODIS DB | 0.91 | 0.86 |

2  **Table 5**. Correlation of annual mean AODs with Fort McMurray AERONET AODs during the respective overlap

3  periods of the various satellite AOD products. In the rightmost column, the contribution of large forest fires has been

4  removed from AERONET data and satellite datasets using +4 standard deviations (σ) as a cutoff.



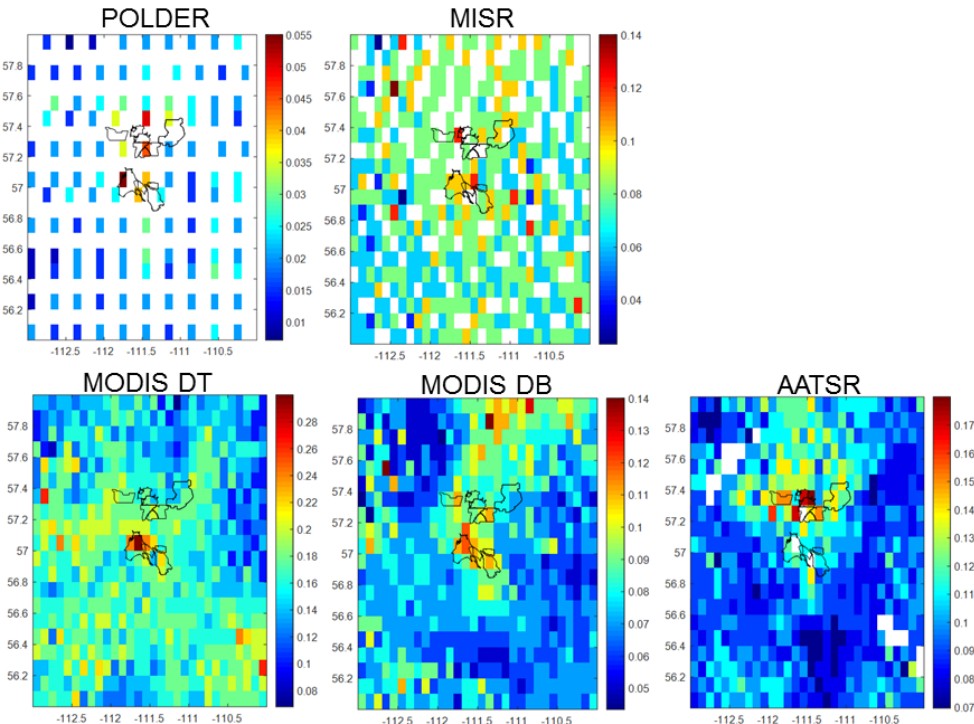

**Figure 1**. Climatological average AOD maps on a 0.1° x 0.1° latitude-longitude grid. (top left) POLDER 865 nm
(1996-2013). Note the gaps in time between the different members of the POLDER series in Table 1. (top right)
MISR 558 nm (2000-2015). (bottom left) MODIS/Aqua DT using only confidence of 3 (2002-2015). (bottom
centre) MODIS/Aqua DB using only confidence of 3 (2002-2015). (bottom right) AATSR 550 nm (2002-2012).
Typical N is ~65 for AATSR (see below) and white areas indicate N<20. Black lines trace out the three surface
mining areas in this and subsequent figures.



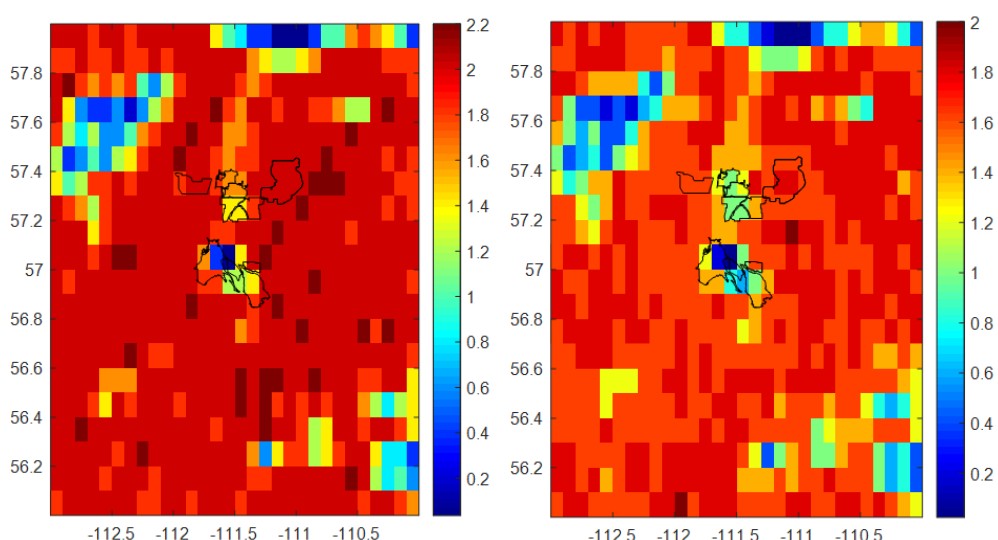

2   **Figure 2**. Map of climatological average confidence (2002-2014) for MODIS/Aqua DT (left) and DB (right) AODs.

3   Lower confidence is expected over Moose Lake (57.6°N, 112.5°W) and the Richardson sand dunes (58.0°N,

4   111.0°W).



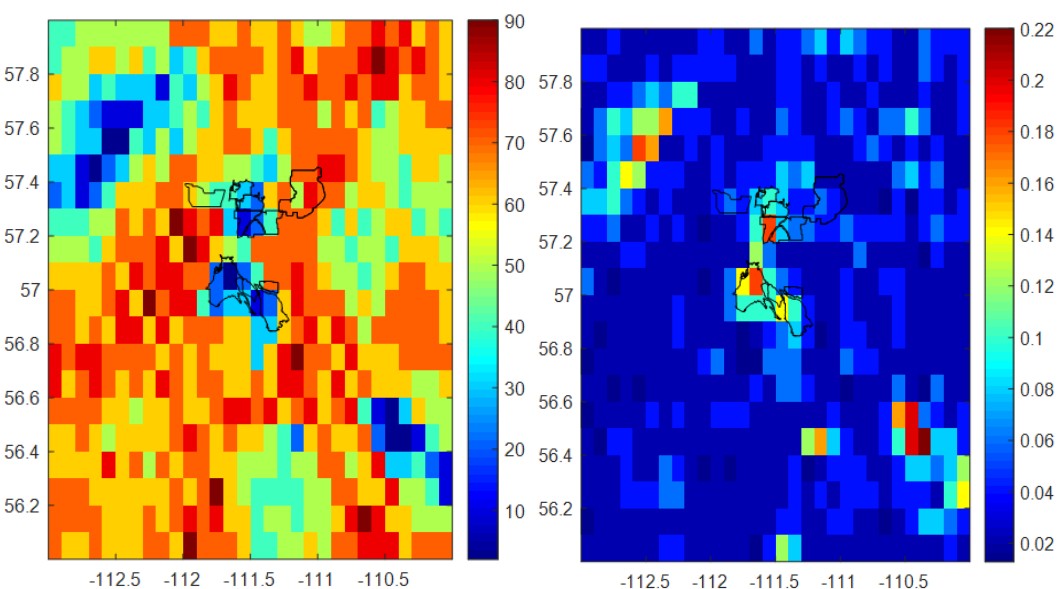

2   **Figure 3**. Map of sample size (left) and average cloud fraction within AOD superpixels when the AOD retrieval is

3   successful (right), compiled from the entire AATSR data record. Smaller sample sizes are expected over Moose

4   Lake and Gordon Lake (56.5°N, 110.5°W).



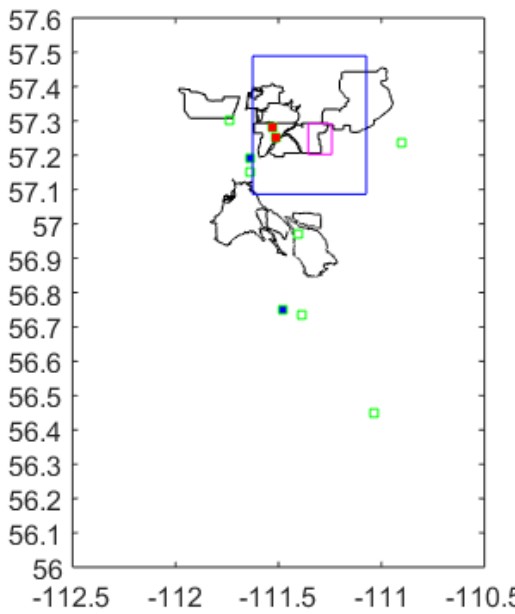

**Figure 4**. Areas with a significant positive trend in AOD in the POLDER/PARASOL, and MODIS/Aqua DB data
records. The area over which the AOD time series is determined for MODIS/Aqua DB (0.1×0.1°), and
POLDER/PARASOL (0.4×0.4°) is outlined in pink and blue, respectively. Locations of 10 NAPS PM$_{2.5}$ monitoring
sites are also shown as small green squares. The central one of 3 adjacent (overlapping) grid cells at constant latitude
is plotted for POLDER/PARASOL (see Sect. 3 for details). The grid cell with the largest trend in the domain is
plotted for MODIS/Aqua DB (see Sect. 3 for details). Note that the Albian mine site (57.2808°N, 111.526°W) was
replaced by the nearby Shell Muskeg River site (57.2491°N, 111.509°W) in 2009 (both station symbols are filled in
red). The two AERONET instruments are co-located with NAPS monitors and those sites are filled in blue.





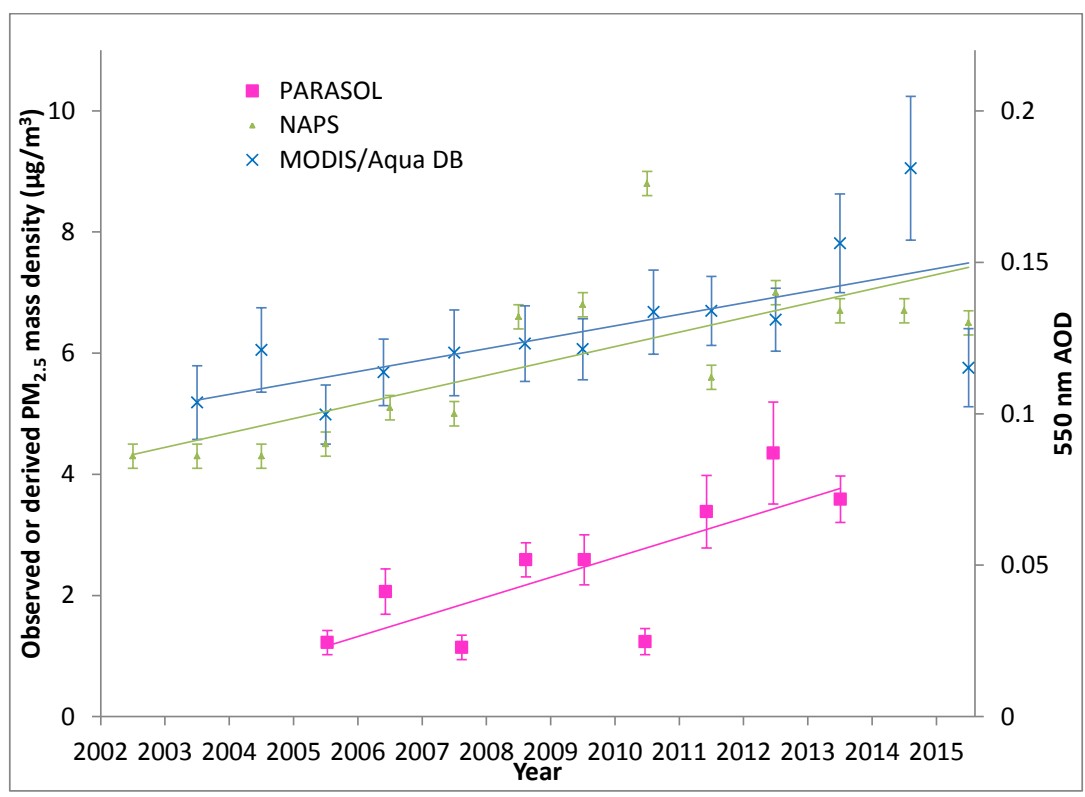

**Figure 5.** Annual average PM$_{2.5}$ mass density for the merged Albian mine and Shell Muskeg River dataset, along
with PM$_{2.5}$ annual averages derived from satellite AOD data records (see Sect. 3 for details and Fig. 4 for satellite
trend areas). Each satellite time series is plotted at the average decimal time for each calendar year. Trend lines are
fitted to each time series using a matching colour. Vertical error bars indicate ±1 standard error of the annual mean.
There are, on average, 33 and 50 observations per year for POLDER/PARASOL and MODIS/Aqua DB,
respectively. The secondary ordinate applies to the MODIS DB observations, but not POLDER/PARASOL (for
which the 865 nm AODs are in the 0.01 to 0.03 range).





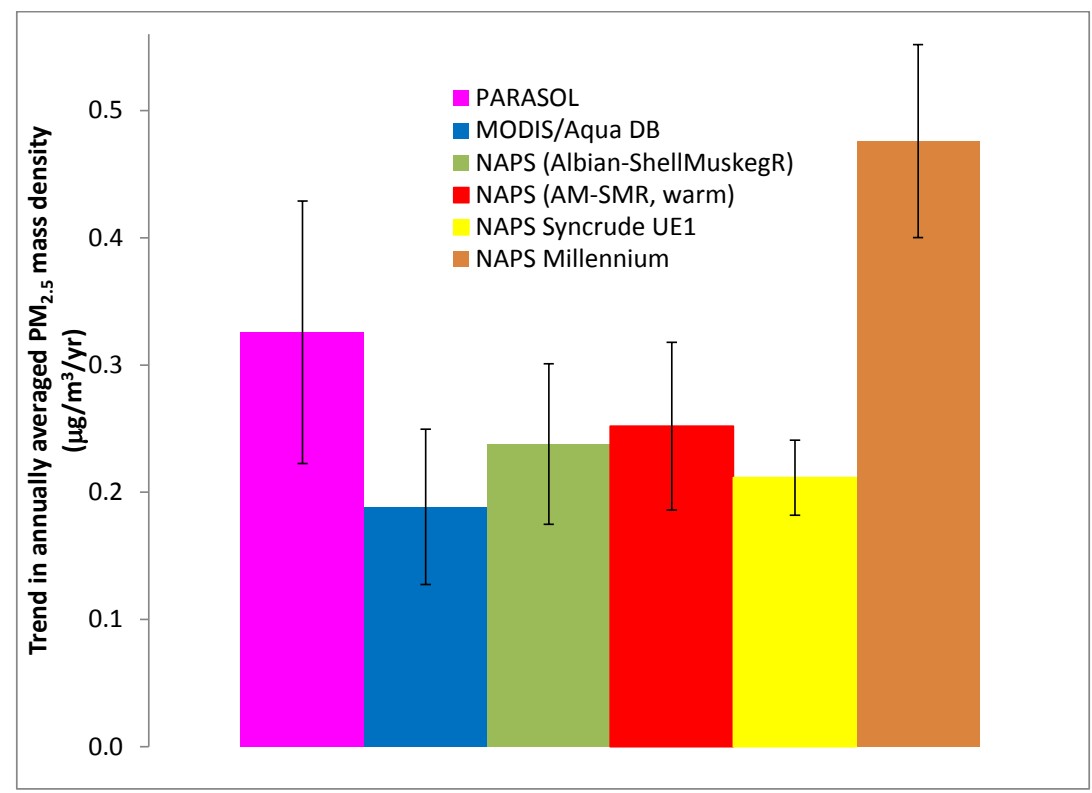

**Figure 6.** Trend in annually averaged $PM_{2.5}$ mass density calculated using NAPS $PM_{2.5}$ data for three locations,
namely the merged Albian mine and Shell Muskeg River dataset (2002-2015), Millennium mine (2002-2014) and
Syncrude UE1 (2003-2014), or derived from satellite AODs in the vicinity of Shell's Albian and Muskeg River
mines (see Fig. 4 and Sect. 3). The trend is also determined for the NAPS $PM_{2.5}$ merged Albian Mine – Shell
Muskeg River (AM-SMR) dataset limiting to the warm season (April to October). Trend uncertainty is indicated
with a vertical bar (±1 s. e.).