# Peer review of "Assessment of the aerosol optical depths measured by satellite- # 2 based passive remote sensors in the Alberta oil sands region"

_Atmospheric Chemistry and Physics, 2016_

## Short Comment (SC1) · 14 Oct 2016

I read this paper with interest, as I am involved with the development of the Deep Blue aerosol products and was recently a co-author on an analysis in this region (Li et al 2016), which the authors have also cited. I had some general comments and suggestions about the use of the satellite aerosol data products used, mostly MODIS. This should not be taken as a full peer-review (I'm not commenting on the PM parts). This is just a comment on the satellite data and AOD analysis.

I am glad to see that the authors used both the Dark Target and Deep Blue products

when conducting their study. However, I see they use the Dark Target AOD product at 470 nm, rather than 550 nm. 550 nm is the main reference wavelength for this product, the one that has been validated, and the one which is generally recommended to be used (and is indeed used by most data users). Keeping everything consistent at 550 nm (or a close wavelength, e.g. MISR's 558 nm) where possible also makes it a clearer comparison between the various products. My suggestion would be to do the analysis with the standard 550 nm AOD product. The Dark Target 550 nm AOD is also contained in the same multidimensional SDS that the authors used (Corrected_Optical_Depth_Land), or the authors may go directly to the combined Dark Target land and ocean SDS Optical_Depth_Land_And_Ocean, which contains AOD at 550 nm with the quality flags already applied (plus includes over-water retrievals). So no additional data download should be necessary to do this. I don't see any reason to actively choose the 470 nm AOD over the standard 550 nm for this analysis.

Similarly, the Deep Blue AOD quality flag is in Deep_Blue_Aerosol_Optical_Depth_550_Land_QA_Flag, but we also provide a data set which already has the quality flag mask applied (Deep_Blue_Aerosol_Optical_Depth_550_Land_Best_Estimate) so the user does not have to do the filtering themselves. It is not clear to me from the paper which SDS was used to QA-filter the Deep Blue data but I am assuming it is the above. More information can be found in the MODIS aerosol file spec document (http://modis-atmos.gsfc.nasa.gov/_specs_c6/MOD04_L2_CDL_2013_03_21.txt) or on our website, http://deepblue.gsfc.nasa.gov. Could this be clarified?

In terms of Aqua vs. Terra, Aqua has indeed been historically more stable and better-characterised so I agree that it is probably sufficient to use only Aqua for the analysis. Note that for Deep Blue we apply additional calibration updates to Terra which are not included by the Dark Target team in the present version of the processing (see Sayer et al JGR, 2015, doi:10.1002/2015JD023878), so our Terra/Aqua differences are smaller than they see and we do not have the same divergence in time through

most of the mission (although in the past year or so there appear to be troubles with Terra calibration again, which are under investigation).

Also, which ATSR product is used? There are at least 3 being produced in Europe in the framework of the ESA CCI project, and they all have different approaches and results (see Popp et al, Remote Sensing, 2016, doi:10.3390/rs8050421 for an overview). My inference is that this is the Swansea algorithm (Peter North's group) but I think this should be stated more clearly. Perhaps the others could be added to the analysis as well, if this is not too much effort. Similar to Dark Target vs. Deep Blue for MODIS, the various ATSR algorithms have different coverage.

For POLDER, the data product the authors have used reports AOD at 865 nm. Due to the wavelength dependence of AOD, in most cases this means that the AOD will be much lower at 865 nm than 550 nm. The smaller signal will probably cause problems for relationships constructed using this AOD, plus one would not expect a close match between AOD at 550 nm (given by the other sensors) and 865 nm since the spectral dependence of AOD is determined by the aerosol composition. I wonder if another POLDER data product like GRASP (see e.g. http://www.grasp-open.com/products/ ) which does report AOD at 550 nm would be more useful here (and also allow for a more direct comparison between the various data sets).

I had also been under the impression that the particular POLDER AOD retrieval data set the authors are using is intended to be only a fine-mode AOD retrieval, rather than a total-AOD retrieval, which further complicates things. However, I may be mistaken about that as I have not used POLDER data myself for a few years now.

I note in the text that AERONET AOD was interpolated to the satellite wavelengths (which is the standard practice), but Table 4's caption says that AERONET data at 500 nm were used. I guess that this is an error in the caption, but can this be clarified?

Figure 1: My guess is that the white spots on the maps for POLDER and MISR are because the level 2 retrievals are at a coarser resolution than the 0.1 degree grid being

[Figure]

averaged to. In that case it might be better to allow the level 2 data to occupy multiple grid cells (corresponding to the actual retrieval footprint) than to snap them to the grid cell nearest to the pixel centre (which is what I assume is being done here). If the retrieval pixels are larger than the grid size (which is the case here) then it does not really make sense to assign a pixel to one grid cell, when it occupies multiple grid cells.

As a general comment on this figure, I would recommend keeping the colour scales the same (and ideally start at zero) to allow a direct comparison between the different data products. Right now it is hard to compare them because the colour bars are different. I realise POLDER is the odd one out here since it is at a longer wavelength, but the other data sets (at or near 550 nm) should be on a consistent scale. I'd also suggest mentioning again in the caption that POLDER is at 865 nm, hence the lower AODs.

As another general comment on the above figure: we know there is seasonal variation in AOD, as well as variation in things that affect sampling (e.g. cloud and snow cover). So presenting an annual mean here conflates these issues together with the issue of retrieval uncertainty. My suggestion would be to make separate maps for each season. They don't all necessarily need to be included in the paper if length is a concern. This way the seasonal aspect at least can be removed and it may bring the different data sets into closer agreement (or it might not). The next stage would be to compare the points only where they have common retrievals on the same days, but I suspect that due to the large number of data sets there would probably be few mutual points. So, making seasonal means rather than annual means is probably a good balance in terms of seeing how the data look compared to each other.

Figure 2: If I understand correctly, this is the mean of the MODIS Deep Blue and Dark Target QA values. I understand the intent behind this figure (illustrate where the algorithms have confidence) but I think the execution is problematic. By taking the mean of the QA flag, it is being treated as a quantitative variable. However it is not – it is a categorical variable that is stored as an integer because it is easy to store integers in the hdf files. QA=0 has a fundamentally different meaning (no retrieval)

from the other values, and the QA from 1 to 3 does not represent linear progression in terms of quantitative retrieval quality or uncertainty. So, taking the mean value is a bit misleading since it is conflating lack of retrievals (due to e.g. clouds) with other algorithm factors and giving a number as a mean for the grid cell which doesn't really relate to the underlying QA flags. For example if the mean QA calculated in this way is 1, it does not mean that the retrievals here have low confidence. It means either that the retrievals have low confidence, or that there is some combination of high confidence retrievals and data gaps due to clouds, etc.

So, I think this figure should be updated, and we might get some more insight into what is going on if the metric here is calculated differently. In Deep Blue we recommend QA=2 and QA=3 can both be used for quantitative analyses as they have similar error characteristics (Sayer et al., JGR 2013, doi: 10.1002/jgrd.50600) while for Dark Target land retrievals they recommend QA=3 only (e.g. Levy et al, ACP 2010, doi:10.5194/acp-10-10399-2010). This is another example of the fact that QA flags have different specific meanings for different data products. What I would suggest is making maps showing the fraction of overpasses where there is no retrieval (i.e. QA=0), the fraction where there is a poor-QA retrieval (i.e. QA=1 for Deep Blue, QA=1 or 2 for Dark Target), and the fraction where there is a good-QA retrieval (i.e. QA=2 or 3 for Deep Blue, QA=3 for Dark Target). Those maps would reveal those areas where the retrievals frequently fail or are absent in a more meaningful way than the current Figure 2, in my view.

Some of the data holes in the MODIS Dark Target product will be from the fact that neither their land nor ocean algorithms treat pixels which are identified as 'coastal' as valid for AOD retrieval. (Note that Deep Blue treats such pixels as land, but excludes pixels next to water frequently for other reasons.) This limits coverage in many parts of Canada and elsewhere in the world, as pixels containing lake shores are frequently identified as coastal. See Carroll et al. (IJDE, 2016, doi: 10.1080/17538947.2016.1232756).

Figure 3: This shows that in areas where there are few AATSR retrievals, those re-trievals that are performed tend to have a higher sub-pixel cloud fraction. The impli-cation is that sampling in this area is influenced by cloud cover, whether real cloud or misidentified cloud (which is reasonable). However what might make a better right panel would be the cloud fraction for ALL observations, not just for those observa-tions where an AOD retrieval is performed. This would look more directly at where the AATSR algorithm thinks there is a cloud. Right now what the panel is showing is subtly different since pixels which are cloudy above the threshold for retrieval (I am not sure if this is 100% cloudy or some lower fraction) are exclude from the analysis.

Table 1: Again, the MODIS standard AOD wavelengths for both Deep Blue and Dark Target are 550 nm. Deep Blue also provides 412, 470, and 650 nm and Dark Target also provides 470 and 650 nm. Source radiances are not all at 0.5 km pixel sizes, it depends on band, so it would be better to say 0.25-1 km here. Also, due to its scan design and wide swath with, MODIS level 1 and level 2 pixel size and shape get heavily distorted from nadir to scan edge (quoted values are all for nadir pixels), which is not an issue for AATSR or MISR to the same degree due to their designs and narrower swaths. See e.g. Sayer et al (AMT, 2015, doi:10.5194/amt-8-5277-2015) for more information.

Table 4 and discussion: I would delete the analysis of linear least-squares regressions from the table and discussion. AOD data violate most/all the assumptions required for this technique to be valid, and so the results are misleading and fits/confidence en-velopes are quantitatively incorrect. See e.g. http://people.duke.edu/~rnau/testing.htm for more discussion. (I know it is a frequently-used technique in our community, but it is fundamentally incorrect for this particular application.)

I hope these comments are useful; please feel free to get in touch if you have questions about them, or about the MODIS aerosol products in general.

---

## Referee Comment (RC1) · Anonymous Referee #1 · 8 Nov 2016

As the title suggests, this paper compares aerosol optical depth (AOD) retrieved by MISR, MODIS, POLDER, and AATSR over the Alberta oil sands region (AOSR). Additionally, using surface based PM2.5 measured by 10 National Air Pollution Surveillance (NAPS) stations, the authors model yearly averaged PM2.5 (and trends) for the various satellite sensors by correlating AOD to PM2.5 for the 10 NAPS sites. Their results indicate that MODIS Deep Blue retrieved AOD and NAPS PM2.5 (at multiple sites) have increased from 2003 through 2014 in a region located within the AOSR. This work is within the scope of ACP and is appropriate for publication after major revisions are made.

Major Comments:

As Andrew Sayer is an expert on aerosol retrievals from satellite-based remote sensing, I strongly recommend the authors fully take his suggestions.

Kahn et al., 2005 describes validation of a previous version of the algorithm and should be replaced with Kahn et al., 2010. The title is "Multiangle Imaging SpectroRadiometer global aerosol product assessment by comparison with the Aerosol Robotic Network". Particle mixtures have changed, but many of the notes the authors have made about MISR remain valid.

Although the paper is focused on AOD trends from satellite-remote sensing, I would recommend also including an analysis of the Fort McMurray AERONET site as well.

Page 8, Line 18-19: The higher SNR is probably irrelevant over land (especially bright surfaces).

PM2.5 Assessment:

I strongly recommend that the authors remove the AOD-to-PM2.5 aspect of this paper. I don't think it adds much to the paper, as the authors have in-situ PM2.5 data for 10 sites anyways, and the correlation between AERONET AOD and satellite remote sensing retrieved AOD is much higher than the correlation between NAPS PM2.5 and satellite remote sensing retrieved AOD. There are also a lot of caveats to converting between an integrated aerosol retrieval (AOD) and a surface aerosol retrieval (PM2.5), many of which I don't see discussed (please correct me if I missed it). Here are some of them:

1. For instance, MISR is viewing this area of the planet at roughly 10:15 AM local time. It is possible that the planetary boundary layer (PBL) is not always fully developed at this time, which would mean that a comparison between MISR AOD and surface based PM2.5 would not be possible.

2. Unmasked transported smoke that happens to be lofted above the PBL may not be

seen by NAPS.

3. Variation in the PBL height from day to day and season to season will cause discrepancies between retrieved AOD and measured PM2.5 using a static ratio.

4. Large-scale differences in land-surface/water coverage may cause systematic discrepancies in PBL height at individual stations.

Although the results of the AOD-to-PM2.5 analysis show a positive trend in PM2.5 from space, I don't really see how useful this is, as the same thing can be shown from the 10 NAPS instruments with a much higher degree of confidence. Additionally, while I may trust the day-to-day changes in AOD retrieved from space, I would never put that kind of faith in converting AOD to PM2.5 on a daily basis. I recognize that the authors did not do this and are basically only using PM2.5 from AOD for yearly analysis, but some people may take this work and try to expand it in ways that probably shouldn't be done.

General Comments:

Is it possible that the drop in 550 nm AOD (Figure 5) and NAPS PM 2.5 during 2015 is related to the fall in oil prices affecting activity in the region? If so, it may be worthwhile to note, as this would likely continue to the present day.

Figure 1: Figure 1 could be improved in a number of ways. In addition to what Andrew Sayer suggested, I recommend putting the locations of your AERONET sites and NAPS stations on the map (maybe as circles and stars). If you wanted to make the plots even more useful, you could color the circles and stars using the same color scale for AERONET, and a different scale for PM2.5.

Figure 5: The authors should include the Fort McMurray AERONET site on this plot as well.

---

## Referee Comment (RC2) · Anonymous Referee #2 · 10 Nov 2016

The manuscript assessed several satellite- based AOD retrievals (POLDER, MISR, AATSR, and MODIS) in the Alberta oil sands region (AOSR) by using two local AERONET sites and several National Air Pollution Surveillance stations. It is within the scope of this journal and in general well written. However, I am concerns that this manuscript is insufficient to be useful due to lack of substantial materials and logical reasoning in current version.

First of all, I have read the comments from Andrew Sayer, who is an expert on aerosol retrievals from satellite-based remote sensing, especially in MODIS AOD retrievals. His

comments are very useful to improve the understanding of the MODIS AOD retrievals and improve current studies.

My major concerns about this manuscript are the lack of in-depth analysis and lack the necessary explanations. For example, the finding of the ability to capture spatial variability with MISR is generally much worse than the other instruments over AOSR region is very interesting and useful to know the limitation of MISR measurements, however the possible reasons for this will be more important to see the spatial limitation of MISR. In section 3.1, the authors have indicated that all of the satellite retrievals can capture the inter-annual variability of the annual mean AOD observed by AERONET, but the trends estimated based on the each satellite retrievals showed lots of differences, some of positive and some of negative. Thus, what are the main reasons to explain this discrepancy? The authors reported a major issue of satellite AOD retrievals over this region, which is the lack of successful retrieval samples, especially of the MODIS retrievals which has low confidence. It is good information. However, the reasons for the large part of retrievals has low confidence are not well explained. Furthermore, the comparison of coincident AODs observed by satellite-based and AERONET shows large bias (more than 20%) between them, but necessary explanations are not provided.

I found that the correlation between monthly mean of the satellite retrieved AOD and AERONET AOD are analyzed, but I'd suggest to use the individual samples from AERONET to evaluate the satellite AOD retrievals and discuss the bias of each satellite product.

It is not clear to describe how to derive the PM2.5 mass density from satellite AODs. I noticed that the constant ratio of PM2.5 to AOD is used to convert the AOD trends from satellite instruments to PM2.5 trends. However, this is not accurate. The relationship between surface PM2.5 and AOD is not always linear. It is affected by multiple factors, such as the relative humidity since the AOD can be enhanced by aerosol swelling effects but the PM2.5 does not. Meanwhile, the correlation between AOD and surface

level PM2.5 significantly depends on the aerosol vertical distribution and aerosol particle size distribution. Thus, the uncertainties in those analysis and the influences on the results should be discussed.

P6, Line 28: Is this trend statistical significant?

---

## Author Comment (AC1) · 17 Jan 2017

Response to short comment by Dr. Andrew Sayer

We thank Dr. Sayer for his suggestions (in red).

However, I see they use the Dark Target AOD product at 470 nm, rather than 550 nm. 550 nm is the main reference wavelength for this product, the one that has been validated, and the one which is generally recommended to be used (and is indeed used by most data users).

We agree with this comment. It was realized after the original submission of the manuscript that the 470 nm product was selected unintentionally instead of the 550 nm product. Rather than withdraw the manuscript or ask for a long extension to regenerate a decade of MATLAB *.mat files required as input for our validation and mapping software, the Dark Target AOD at 470 nm was retained temporarily with the full intention of redoing the map in Fig. 1, the validation results (Table 4), *et cetera*, at the next stage in the review process.

We now write at p2L28:

Specifically, the Corrected_Optical_Depth_Land (550 nm) and the Deep_Blue_Aerosol_Optical_Depth_550_Land datasets were used and confidence for both datasets was extracted from the Quality_Assurance_Land dataset.

Similarly, the Deep Blue AOD quality flag is in Deep_Blue_Aerosol_Optical_Depth_550_Land_QA_Flag, but we also provide a data set which already has the quality flag mask applied (Deep_Blue_Aerosol_Optical_Depth_550_Land_Best_Estimate) so the user does not have to do the filtering themselves. It is not clear to me from the paper which SDS was used to QA-filter the Deep Blue data but I am assuming it is the above. More information can be found in the MODIS aerosol file spec document (http://modisatmos.gsfc.nasa.gov/_specs_c6/MOD04_L2_CDL_2013_03_21.txt) or on our website, http://deepblue.gsfc.nasa.gov. Could this be clarified?

We agree that the ACPD manuscript fails to name the SDS used to QA-filter the Deep Blue data. 'Quality_Assurance_Land' is the SDS used.

The change to the manuscript is contained in the sentence mentioned above at p2L28, in response to the previous comment.

Also, which ATSR product is used? There are at least 3 being produced in Europe in the framework of the ESA CCI project, and they all have different approaches and results (see Popp et al, Remote Sensing, 2016, doi:10.3390/rs8050421 for an overview). My inference is that this is the Swansea algorithm (Peter North's group) but I think this should be stated more clearly.

The selected ATSR product is stated clearly in the appendix of the existing manuscript (p13L17) and the appendix is referenced at p2L26 in connection with the satellite data products. Information on the ATSR product is in the following sentence of the appendix (p13L17)

AATSR and ATSR-2 version 4.1 data are from Swansea University and can be obtained from the Aerosol CCI website (http://www.esa-aerosol-cci.org/) following registration.

Perhaps the others could be added to the analysis as well, if this is not too much effort. Similar to Dark Target vs. Deep Blue for MODIS, the various ATSR algorithms have different coverage.

There are three different algorithms for both AATSR and ATSR-2, and at least two POLDER algorithms, several MODIS products (Terra vs. Aqua, Deep Blue vs. Dark Target), plus MISR. That is eleven, and it is not an exhaustive list of available products from these satellite-based sensors. The primary focus of this paper is not on algorithms but on the different aerosol sensors. The Swansea University algorithm was chosen since initially they had, by far, the longest AATSR data record available.
To make this decision clear, we now write at p3L9:

The focus in this paper is primarily on the different aerosol sensors, rather than the different retrieval algorithms applied to the same satellite data (e.g. Popp et al., 2016), with the exception of the widely used Deep Blue and Dark Target algorithms for MODIS.

For POLDER, the data product the authors have used reports AOD at 865 nm. Due to the wavelength dependence of AOD, in most cases this means that the AOD will be much lower at 865 nm than 550 nm. The smaller signal will probably cause problems for relationships constructed using this AOD, plus one would not expect a close match between AOD at 550 nm (given by the other sensors) and 865 nm since the spectral dependence of AOD is determined by the aerosol composition. I wonder if another POLDER data product like GRASP (see e.g. http://www.grasp-open.com/products/) which does report AOD at 550 nm would be more useful here (and also allow for a more direct comparison between the various data sets).

We have used the only POLDER AOD data product that was available at CNES's POLDER website. We did not search the web or the literature for alternate POLDER products.

The different satellite AOD data sets are essentially not compared in a quantitative way. The quantitative comparison is essentially against AERONET and thus the different wavelength (865 versus 550 nm) is not a major issue since AERONET measures at 870 nm and many wavelengths in the visible. The smaller aerosol signal at 865 nm does not cause problems for the linear regression relationship constructed between POLDER and AERONET AODs. This is obvious from the high correlation coefficients for POLDER in Tables 3-5. Also POLDER reports AOD at 865 nm, but uses measurements at 670 nm in the AOD retrieval.

I had also been under the impression that the particular POLDER AOD retrieval data set the authors are using is intended to be only a fine-mode AOD retrieval, rather than a total-AOD retrieval, which further complicates things. However, I may be mistaken about that as I have not used POLDER data myself for a few years now.

Dr. Sayer makes an interesting point here. This is not a fine-mode AOD product; total AOD is retrieved and reported. See:
 http://www.icare.univ-lille1.fr//projects_data/parasol/docs/Parasol_Level-2_format.pdf.
However, the use of polarized radiances in the POLDER retrieval greatly reduces the sensitivity of the retrieval to coarse particles. Thus, it is possible that a coarse-mode aerosol plume could, to some extent, mask the polarization signal from underlying fine-mode particles if such an arrangement occurred. Ultimately, the low sensitivity of POLDER to coarse-mode particles appears to be a minor issue at the two AERONET sites (Fort McMurray and Fort McKay) given the lack of bias and the high degree of correlation with AERONET AOD, in spite of the fact that coarse-mode dust is known to be significant contributor in this region, particularly at Fort McKay (based on AATSR dust fraction, not shown).

I note in the text that AERONET AOD was interpolated to the satellite wavelengths (which is the standard practice), but Table 4's caption says that AERONET data at 500 nm were used. I guess that this is an error in the caption, but can this be clarified?

Dr. Sayer is correct that this needs clarification, even though there is not an error. AERONET 500 nm AOD is used, however it is scaled to the satellite wavelengths.
In the caption, we now write:

The Cimel 500 nm AOD, scaled to the satellite AOD wavelength (see Sect. 2), is used for comparison with all satellite sensors except POLDER/PARASOL, for which the Cimel 870 nm AOD is more appropriate (see Table 1).

In that case it might be better to allow the level 2 data to occupy multiple grid cells (corresponding to the actual retrieval footprint) than to snap them to the grid cell nearest to the pixel centre (which is what I assume is being done here). If the retrieval pixels are larger than the grid size (which is the case here) then it does not really make sense to assign a pixel to one grid cell, when it occupies multiple grid cells.

The orientation of actual footprint would need to be known and, for POLDER, this information is not available for each observation: only the latitude and longitude at the center of the AOD superpixel is provided. In general, we disagree that it does not make sense to assign a pixel to one grid cell. This is referred to as spatial oversampling and can be very revealing about localized sources of aerosols.

As a general comment on this figure, I would recommend keeping the colour scales the same (and ideally start at zero) to allow a direct comparison between the different data products. Right now it is hard to compare them because the colour bars are different. I realise POLDER is the odd one out here since it is at a longer wavelength, but the other data sets (at or near 550 nm) should be on a consistent scale. I'd also suggest mentioning again in the caption that POLDER is at 865 nm, hence the lower AODs.

This recommendation initially seemed like a good one, but even the AOD differences between the MODIS products using the respective confidence values suggested by Dr. Sayer near the Syncrude facility are quite large, as shown in this Deep Blue climatological mean AOD map using confidence 2-3, but with the AOD range of the colour bar extending to 0.26 to cover the maximum climatological AOD of Dark Target (confidence=3).

[Figure]

Including such a figure would severely compromise our primary goal for Fig. 1, which is to show the spatial gradients in AOD in this region. The colour scales have been changed to have a common lower limit of 0.

We already mentioned in the caption that POLDER is at 865 nm: "(top left) POLDER 865 nm (1996-2013)". Just as a point of information, the Deep Blue climatological AOD for confidence=3 has a hotspot near the Suncrude facility with AOD of 0.12, yet we find that higher climatological maximum AODs occur (0.18) when only confidences of 1-2 are retained, again with the hotspot being the Syncrude Mildred Lake facility, as shown in the following maps to the left and right, respectively.

[Figure]

As another general comment on the above figure: we know there is seasonal variation in AOD, as well as variation in things that affect sampling (e.g. cloud and snow cover). So presenting an annual mean here conflates these issues together with the issue of retrieval uncertainty. My suggestion would be to make separate maps for each season. They don't all necessarily need to be included in the paper if length is a concern. This way the seasonal aspect at least can be removed and it may bring the different data sets into closer agreement (or it might not). The next stage would be to compare the points only where they have common retrievals on the same days, but I suspect that due to the large number of data sets there would probably be few mutual points. So, making seasonal means rather than annual means is probably a good balance in terms of seeing how the data look compared to each other.

We tried plotting AODs for May through September for the MODIS and MISR products (Figures A-C below). These are the months when all five aerosol products have high measurement frequency. But again, Dr. Sayer's purpose is evidently different than ours: we are not trying to bring the different data sets into closer agreement; as stated up front (p4L33), we are mostly trying to see what each is capturing spatially over the long term, so annual means are preferable. Anyway, as shown in Figures A-C below, limiting to these 'warm season' months does not bring the data sets into closer agreement. Limiting to the warm season was mostly expected to benefit the MISR AOD map since MISR has an unusual spatiotemporal sampling pattern, but as shown in Figure C, limiting to May-September does not produce a more coherent AOD map. In the revised manuscript, all available months are retained for Figure 1.

[Figure]

Figure A: MODIS DT 550 nm climatological AOD for May to September (confidence=3).

[Figure]

Figure B: MODIS DB 550 nm climatological AOD for May to September for confidence≥2.

[Figure]

Figure C: MISR 558 nm climatological AOD for May to September.

Figure 2: If I understand correctly, this is the mean of the MODIS Deep Blue and Dark Target QA values. I understand the intent behind this figure (illustrate where the algorithms have confidence) but I think the execution is problematic. By taking the mean of the QA flag, it is being treated as a quantitative variable. However it is not – it is a categorical variable that is stored as an integer because it is easy to store integers in the hdf files. QA=0 has a fundamentally different meaning (no retrieval) from the other values, and the QA from 1 to 3 does not represent linear progression in terms of quantitative retrieval quality or uncertainty. So, taking the mean value is a bit misleading since it is conflating lack of retrievals (due to e.g. clouds) with other algorithm factors and giving a number as a mean for the grid cell which doesn't really relate to the underlying QA flags. For example if the mean QA calculated in this way is 1, it does not mean that the retrievals here have low confidence. It means either that the retrievals have low confidence, or that there is some combination of high confidence retrievals and data gaps due to clouds, etc.

This comment by Dr. Sayer is correct, and we were aware of all of these logical points. The main purpose of both panels of Fig. 2 was to show that QA is tending very close to 0 (i.e. <0.45) at the two grid cells near the Syncrude Mildred Lake facility, implying that the retrieval has no confidence (or provides a fill value) more than 55% of the time.

So, I think this figure should be updated, and we might get some more insight into what is going on if the metric here is calculated differently. In Deep Blue we recommend QA=2 and QA=3 can both be used for quantitative analyses as they have similar error characteristics (Sayer et al., JGR 2013, doi: 10.1002/jgrd.50600) while for Dark Target land retrievals they recommend QA=3 only (e.g. Levy et al, ACP 2010, doi:10.5194/acp-10-10399-2010). This is another example of the fact that QA flags

have different specific meanings for different data products. What I would suggest is making maps showing the fraction of overpasses where there is no retrieval (i.e. QA=0), the fraction where there is a poor-QA retrieval (i.e. QA=1 for Deep Blue, QA=1 or 2 for Dark Target), and the fraction where there is a good-QA retrieval (i.e. QA=2 or 3 for Deep Blue, QA=3 for Dark Target).

This suggestion is accepted. A new six-panel Fig. 2 has been generated.

Some of the data holes in the MODIS Dark Target product will be from the fact that neither their land nor ocean algorithms treat pixels which are identified as 'coastal' as valid for AOD retrieval. (Note that Deep Blue treats such pixels as land, but excludes pixels next to water frequently for other reasons.) This limits coverage in many parts of Canada and elsewhere in the world, as pixels containing lake shores are frequently identified as coastal. See Carroll et al. (IJDE, 2016, doi: 10.1080/17538947.2016.1232756).

This cause of data holes has been added to the list of causes. We now write at p5L16:

The number of pixels used in the AOD retrieval is reduced by the inland water mask (Carroll et al., 2016), …

Figure 3: This shows that in areas where there are few AATSR retrievals, those retrievals that are performed tend to have a higher sub-pixel cloud fraction. The implication is that sampling in this area is influenced by cloud cover, whether real cloud or misidentified cloud (which is reasonable). However what might make a better right panel would be the cloud fraction for ALL observations, not just for those observations where an AOD retrieval is performed. This would look more directly at where the AATSR algorithm thinks there is a cloud. Right now what the panel is showing is subtly different since pixels which are cloudy above the threshold for retrieval (I am not sure if this is 100% cloudy or some lower fraction) are exclude from the analysis.

Additional cloud tests (Bevan et al., 2012 and reference therein) were used for this AATSR aerosol retrieval algorithm that are not used in AATSR Instrument Processing Facility (IPF) v6.01 cloud product. Thus, we feel it is more appropriate to look at the cloud fractions in the successful AOD retrievals. This suggestion might have been worth pursuing if the spatial anti-correlation was not strong between cloud fraction in successful AOD retrievals and AOD sample size, but that is not the case.

Table 1: Again, the MODIS standard AOD wavelengths for both Deep Blue and Dark Target are 550 nm. Deep Blue also provides 412, 470, and 650 nm and Dark Target also provides 470 and 650 nm. Source radiances are not all at 0.5 km pixel sizes, it depends on band, so it would be better to say 0.25-1 km here. Also, due to its scan design and wide swath with, MODIS level 1 and level 2 pixel size and shape get heavily distorted from nadir to scan edge (quoted values are all for nadir pixels), which is not an issue for AATSR or MISR to the same degree due to their designs and narrower swaths. See e.g. Sayer et al (AMT, 2015, doi:10.5194/amt-8-5277-2015) for more information.

In Table 1, regarding the spatial resolution of MODIS radiances, we now write: $0.25 \times 0.25$ to $1 \times 1$. We have also changed one column heading to: "Spatial resolution of AOD superpixel at nadir".

Table 4 and discussion: I would delete the analysis of linear least-squares regressions from the table and discussion. AOD data violate most/all the assumptions required for this technique to be valid, and so the results are misleading and fits/confidence envelopes are quantitatively incorrect. See e.g. http://people.duke.edu/_rnau/testing.htm for more discussion. (I know it is a frequently-used technique in our community, but it is fundamentally incorrect for this particular application.)

Dr. Sayer's most recent paper (Carroll et al., 2016) cites Levy et al. (2013) for AOD validation, and Dr. Sayer is also a co-author in the latter work. This latter work includes linear least-squares regression of MODIS AOD and AERONET AOD (their Fig. 11), which is precisely what we have done. It is clear that our Table 4 adheres to the established convention in this field in terms of validation statistics. As an alternative, we tested two non-parametric methods (Theil's complete and incomplete methods) to obtain the values in the first three columns of values in Table 4. None of the assumptions are violated when using Theil's incomplete method (1950). Also, application of Spearman's rank correlation is valid for this application (see Table 4). As shown in the table below, the non-parametric methods yielded slopes that were small (~0.6) and ordinary least-squares ('OLS') yielded a slope that was clearly of the wrong sign due to one small cluster of outliers at high AOD. We tested a number of robust regression methods compared in Holland and Welsch (1977), which all use a weighted least-squares (WLS) approach to reduce the sensitivity to anomalous data pairs (i.e. coincidences). Some of these robust regression methods are expected to perform better than OLS on data with non-Gaussian distributions (e.g. Andrews, 1974). The outliers affect whether the AERONET and satellite AOD data conform to a normal distribution. The table below presents the slope and offset from various robust methods using the POLDER/PARASOL and AERONET coincident data at Fort McMurray:

| Method | offset | slope |
|---|---|---|
| Andrews | -0.017 | 0.787 |
| bisquare | -0.017 | 0.788 |
| Cauchy | -0.017 | 0.797 |
| Fair | -0.019 | 0.859 |
| Huber | -0.018 | 0.831 |
| logistic | -0.018 | 0.835 |
| Talwar | -0.017 | 0.787 |
| Welsch | -0.017 | 0.791 |
| OLS | -0.030 | 1.10 |
| Theil's "incomplete" | -0.009 | 0.590 |
| Theil's "complete" | -0.010 | 0.620 |

It is clear that POLDER has a negative offset, but the magnitude of the offset falls into three groups: OLS, robust WLS methods (first eight rows of table above) and robust non-parametric methods. Identical groupings of regression methods are found upon examining the slope values. Furthermore, omitting the small cluster of points with AERONET AOD>0.8, which were all measured on one day, namely 16 July 2012, the OLS slope becomes 0.7904 and offset is -0.014. This slope and offset are both very close to the slope and offset values from the various WLS fits. In the revised manuscript, we select to weight the fit residuals with Huber's function, for the following reason given by Bergstrom and Edlund (2013):

"while it still is robust, it does not completely disregard highly deviating points".

The table above shows that neither the offset nor the slope obtained with the Huber weights are outliers within the WLS group of robust regression methods. The tuning constant is assumed to be 1.345 following Holland and Welsch (1977).
At p3L29 of the revised manuscript, we now write:

Since individual AERONET and satellite AODs are not normally distributed, we use linear least-squares weighted by Huber's function to determine the slope and offset since this is a robust method that does not completely disregard highly deviating points (Bergström and Edlund, 2014). The slope and offset values determined using Huber's weighting function are encompassed by the values obtained with seven alternative weighting functions. Similarly, due to the non-normal distribution of the individual AOD data, Spearman's rank correlation coefficient ($r_s$) is chosen to study the site-specific AOD correlation based on individual AERONET-satellite coincidences.

**References**

Andrews, D. A., A robust method for multiple linear regression, Technometrics, 16(4), 523-531, 1974.

Bergström, P., and Edlund, O.: Robust registration of point sets using iteratively reweighted least squares, Comput. Optim. Appl., 58, 543–561, 2014.

Theil, H.: A rank-invariant method for linear and polynomial regression analysis: I, Proc. Kon. Ned. Akad. Wetensch., 53, 386-392, 1950.

---

## Author Comment (AC2) · 17 Jan 2017

**Response to comments by reviewer 1**

We thank the reviewer for sharing their expertise and improving the manuscript.

Major Comments:
As Andrew Sayer is an expert on aerosol retrievals from satellite-based remote sensing,
I strongly recommend the authors fully take his suggestions.

We agree with this major comment and have taken most of the Dr. Sayer's suggestions.

Kahn et al., 2005 describes validation of a previous version of the algorithm and should
be replaced with Kahn et al., 2010. The title is "Multiangle Imaging SpectroRadiometer
global aerosol product assessment by comparison with the Aerosol Robotic Network".
Particle mixtures have changed, but many of the notes the authors have made about
MISR remain valid.

We have used the more recent reference suggested by the reviewer. We now write in Sect. 4:

The MISR low bias may be related to the need for darker spherical particles (Kahn et al., 2010) given that
forest fire smoke plays a significant role throughout western Canada in the warm season (O'Neill et al.,
2002). Spherical particles with lower single scattering albedo (SSA) may also be required to properly
represent local anthropogenic pollution (Kahn et al., 2010) in the AOSR.

Although the paper is focused on AOD trends from satellite-remote sensing, I would
recommend also including an analysis of the Fort McMurray AERONET site as well.

The AERONET data record is short (2005-2015) at Fort McMurray and includes a missing year (2006)
and three currently incomplete years (2005, 2007, and 2015). The record effectively spans 2008 to 2014,
which is too short for trend analysis, given the large interannual variability.

Page 8, Line 18-19: The higher SNR is probably irrelevant over land (especially bright
surfaces).

Most of the retrieved AODs used in the temporal correlation with the Fort McMurray AERONET site, at
least by MODIS, are over dark vegetation. However, SNR is valuable both for dark and bright scenes. To
first order, the bright surface does not affect the number of detected aerosol-scattered photons, it
essentially affects the number of photons reflected by the surface. So while a bright scene has less noisy
radiances, the fractional contribution by aerosol scattering decreases relative to a dark scene and greater
SNR is required to be able to detect a typical, small AOD (e.g. 0.1 at 550 nm) with comparable AOD
precision relative to a dark scene. In spite of this point, we agree that the SNR of all instruments is
probably sufficient and the higher SNR is likely irrelevant.
Thus, the relevant sentence in the manuscript becomes:

Stronger short-term correlation with AERONET AODs reflects the superior spatial resolution of the MODIS
radiances (Table 1).

PM2.5 Assessment:
I strongly recommend that the authors remove the AOD-to-PM2.5 aspect of this paper.
I don't think it adds much to the paper, as the authors have in-situ PM2.5 data for
10 sites anyways, and the correlation between AERONET AOD and satellite remote
sensing retrieved AOD is much higher than the correlation between NAPS PM2.5 and

satellite remote sensing retrieved AOD.

There are also a lot of caveats to converting between an integrated aerosol retrieval (AOD) and a surface aerosol retrieval (PM2.5), many of which I don't see discussed (please correct me if I missed it). Here are some of them:
1. For instance, MISR is viewing this area of the planet at roughly 10:15 AM local time.
It is possible that the planetary boundary layer (PBL) is not always fully developed at this time, which would mean that a comparison between MISR AOD and surface based
PM2.5 would not be possible.
2. Unmasked transported smoke that happens to be lofted above the PBL may not be seen by NAPS.
3. Variation in the PBL height from day to day and season to season will cause discrepancies between retrieved AOD and measured PM2.5 using a static ratio.
4. Large-scale differences in land-surface/water coverage may cause systematic discrepancies in PBL height at individual stations.
Although the results of the AOD-to-PM2.5 analysis show a positive trend in PM2.5 from space, I don't really see how useful this is, as the same thing can be shown from the 10 NAPS instruments with a much higher degree of confidence. Additionally, while I may trust the day-to-day changes in AOD retrieved from space, I would never put that kind of faith in converting AOD to PM2.5 on a daily basis. I recognize that the authors did not do this and are basically only using PM2.5 from AOD for yearly analysis, but some people may take this work and try to expand it in ways that probably shouldn't be done.

We agree with these comments. The AOD to $PM_{2.5}$ aspect can be avoided with the approach used in the revised manuscript. This involves correcting the POLDER/PARASOL and MODIS Deep Blue offsets (determined from the AERONET validation at Fort McMurray) and then calculating relative trends for AODs (from satellite) and for $PM_{2.5}$ (NAPS).

This is now described at p4L12:

For temporal trends, a simple linear regression is performed on relative anomalies derived from bias-corrected annual average and median AODs. The bias correction involves subtracting the AOD offset obtained through the validation with coincident Fort McMurray AERONET data.

General Comments:
Is it possible that the drop in 550 nm AOD (Figure 5) and NAPS PM 2.5 during 2015 is related to the fall in oil prices affecting activity in the region? If so, it may be worthwhile to note, as this would likely continue to the present day.

It is possible, but not likely, and this is too speculative in our opinion given that NAPS $PM_{2.5}$ data is not significantly different in 2013 and 2015 (see figure below illustrating oil prices over the past seven years). (http://www.nasdaq.com/markets/crude-oil.aspx?timeframe=7y).

[Figure]

Figure 1: Figure 1 could be improved in a number of ways. In addition to what Andrew Sayer suggested, I recommend putting the locations of your AERONET sites and NAPS stations on the map (maybe as circles and stars). If you wanted to make the plots even more useful, you could color the circles and stars using the same color scale for AERONET, and a different scale for PM2.5.

The maps in Fig. 1 use all available satellite data, not just data that is coincident with $PM_{2.5}$ or AERONET observations. $PM_{2.5}$ is measured at night and in winter, when these satellite instruments do not measure. Similarly, the AERONET sites in the oil sands region measure all day, not just at the 1 or 2 local times per day of the satellite instruments and we have found diurnal variations in AOD of 30% at Fort McMurray based on AERONET data. Furthermore, AERONET has slightly more coverage during the cold season. To avoid these biases, in Fig. 1, we plot only the average AOD from satellite-coincident AERONET measurements. Both AERONET sunphotometers in the AOSR are collocated with NAPS sensors, so we chose AERONET over NAPS for Fig. 1. There is also the problem of a possible trend. The NAPS or AERONET data may cover a significantly shorter period (e.g. AERONET at Fort McKay started in 2013 whereas the POLDER map includes data from 1996). We leave Fort McKay out since the data record is too short for a reliable climatology of coincident AODs and has no temporal overlap with most of the sensors.

Figure 5: The authors should include the Fort McMurray AERONET site on this plot as well.

See response to earlier, related comment. No change is made to the manuscript.

---

## Author Comment (AC3) · 17 Jan 2017

**Response to reviewer 2**

We begin by thanking the reviewer for their very helpful comments.

However, I am concerns that this manuscript is insufficient to be useful due to lack of substantial materials and logical reasoning in current version.

We have added explanations to substantiate some of the results. For example, we now discuss a possible cause for why MISR does not capture the hotspot in climatological AOD as well as the other instruments. We have provided a reason why median AOD and $PM_{2.5}$ mass densities are preferable for the spatial correlation analysis in the revised manuscript (as opposed to mean values used in the original manuscript). We feel the discrepancy in long-term trends between the satellite sensors is not strong, but now suggest that the MODIS calibration degradation could account for the general negative trend in AOD from this sensor. Further details are provided below on each issue. This is simply a summary of our response.

First of all, I have read the comments from Andrew Sayer, who is an expert on aerosol retrievals from satellite-based remote sensing, especially in MODIS AOD retrievals. His comments are very useful to improve the understanding of the MODIS AOD retrievals and improve current studies.

Dr. Sayer's comments have helped to improve the revised manuscript. The reviewer can refer to our response to Dr. Sayer's comments to see the resulting changes to the manuscript.

My major concerns about this manuscript are the lack of in-depth analysis and lack the necessary explanations. For example, the finding of the ability to capture spatial variability with MISR is generally much worse than the other instruments over AOSR region is very interesting and useful to know the limitation of MISR measurements, however the possible reasons for this will be more important to see the spatial limitation of MISR.

The MISR spatial limitation, evident in Fig. 1, is probably due to its spatial sampling being tied to its temporal sampling. We found locations within the AOSR where MISR was measuring almost exclusively in October. Thus, the seasonal cycle in AOD is aliasing into the AOD spatial distribution.
The spatial correlation coefficient is based on 10 sites. Because of the small number of sites, the correlation is quite sensitive to a bias in AOD or $PM_{2.5}$ at any station. Wapasu has significantly higher mean $PM_{2.5}$ mass density for MISR coincidences than any other site (10.2 μg/m$^3$ while the next highest site average is 8.1 μg/m$^3$). MISR overpasses of Wapasu span only two years (2014-2015) and these years were affected by anomalously high forest fire activity in western Canada. The median reduces the sensitivity to these outliers as compared to the mean. In the revised manuscript, Table 3 now contains the correlation of the median of coincident $PM_{2.5}$ and satellite AOD data. This table is inserted below. The revised Table 3 shows the spatial correlation coefficient (R) of MISR AOD with $PM_{2.5}$ is not much worse than the spatial R of MODIS/Aqua DT and $PM_{2.5}$.

| AOD product | R | N |
|---|---|---|
| POLDER/PARASOL 865 nm | 0.64 | 8 |
| AATSR 550 nm | 0.73 | 9 |
| MISR 558 nm | 0.20 | 10 |
| MODIS/Aqua DT 550 nm | 0.23 | 10 |
| MODIS/Aqua DB 550 nm | 0.57 | 10 |

At p3L34, we now modify the description of the spatial correlation analysis as follows:

In order to assess the ability of the satellite data to capture the spatial variability in this region, the hourly in-situ surface-level $PM_{2.5}$ from the 10 NAPS (National Air Pollution Surveillance) stations (Table 2) are used. Demerjian (2000) provided a review of the NAPS network, but since 2011, this network has undergone a gradual shift in the continuous monitoring of $PM_{2.5}$ mass density from tapered element oscillating microbalances (TEOMs) to the SHARP (Synchronized Hybrid Ambient Real-time Particulate) monitoring system. The latter is a hybrid system, consisting of a nephelometer and a beta attenuation monitor (Hsu et al., 2016). The spatial correlation between median satellite AODs and NAPS $PM_{2.5}$ mass densities is determined using coincident data.

At p5L4, we now update the text with the following:

The AOD hotspot in the AOSR seen by POLDER is less obvious with MISR (Fig. 1). This is consistent with the relatively poorer ability of MISR to capture spatial variability based on spatial correlations of median AOD and $PM_{2.5}$ mass density over the ~10 available sites (Table 3). While the spatial correlation analysis relies on temporally coincident data, the less obvious AOD hotspot for MISR in Fig. 1 is also partly due to the spatiotemporal sampling by this instrument. Some locations are only sampled during a short period of the year, and thus the seasonal cycle of AOD is aliased into the MISR spatial distribution.

In section 3.1, the authors have indicated that all of the satellite retrievals can capture the inter-annual variability of the annual mean AOD observed by AERONET, but the trends estimated based on the each satellite retrievals showed lots of differences, some of positive and some of negative. Thus, what are the main reasons to explain this discrepancy?

We agree that there is a discrepancy between the trends estimated by the different satellite AOD products, but it is not strong. The satellite data records all span approximately one decade. A period of a decade is rather short for determining a trend, considering the natural interannual variability in AOD and possible instrumental drifts (e.g. Levy et al., 2015). Focussing on the Muskeg River mining region where there appears to be a significant positive AOD trend according to MODIS/Aqua DB and POLDER/PARASOL, the linear trend is not different from zero for both AATSR and MISR (p6L28-29). Also, MODIS/Aqua DT has a slightly negative trend, but it is also not different from a null trend, so given that none of AOD products show a strong decreasing trend in this Muskeg River mining region, there is no strong discrepancy in the AOD trends. The insignificant negative AOD trend for MODIS/Aqua DT remains now that we have switched to 550 nm.

We now add at p8L1:

The calibration of the MODIS reflective solar bands is achieved by calibration with the solar diffuser. Some negative drift in AOD (Levy et al., 2015) is expected for MODIS Aqua similar to its Terra counterpart (see Sect. 2)

as the designs of the solar diffuser and its stability monitor are nearly identical in the two MODIS sensors (Wu et al., 2013).

The authors reported a major issue of satellite AOD retrievals over this region, which is the lack of successful retrieval samples, especially of the MODIS retrievals which has low confidence. It is good information. However, the reasons for the large part of retrievals has low confidence are not well explained.

The reasons for the low confidence of MODIS AODs were explained in the ACPD version of the manuscript (p5L24-26 for Deep Blue and p5L12-19 for Dark Target). An additional reason for MODIS Dark Target has been added to the revised manuscript: coastal areas (see comment by Dr. Sayer and response).

Furthermore, the comparison of coincident AODs observed by satellite-based and AERONET shows large bias (more than 20%) between them, but necessary explanations are not provided.

MISR is the only satellite-based aerosol sensors with a consistent bias of >20% in this region. Explanations were included in the ACPD version (p9L5-11), although one literature reference has been updated in these sentences.

I found that the correlation between monthly mean of the satellite retrieved AOD and AERONET AOD are analyzed, but I'd suggest to use the individual samples from AERONET to evaluate the satellite AOD retrievals and discuss the bias of each satellite product.

This is already done in Table 4. The second to fourth columns in Table 4, namely '$r_s$', 'slope', and 'offset', are all based on individual coincidences. Although it can be inferred from the ACPD version of the manuscript that the quantities in these columns are based on a regression using individual coincidences (e.g. p1L12 and p9L3-4), we will be more explicit in Sect. 2. At p3L30, we now write

"Since individual AERONET and satellite AODs are not normally distributed, we use linear least-squares weighted by Huber's function to determine the slope and offset since this is a robust method that does not completely disregard highly deviating points (Bergström and Edlund, 2014). (…) Similarly, due the non-normal distribution of the individual AOD data, Spearman's rank correlation coefficient ($r_s$) is chosen to study the site-specific AOD correlation based on individual AERONET-satellite coincidences."

In the conclusion (p8L17), we now repeat that correlation was determined using individual AERONET observations:

"However, the MODIS dark target product is the best at capturing temporal variability in terms of the correlation with individual AERONET AODs at Fort McMurray…"

It is not clear to describe how to derive the PM2.5 mass density from satellite AODs. I noticed that the constant ratio of PM2.5 to AOD is used to convert the AOD trends from satellite instruments to PM2.5 trends. However, this is not accurate. The relationship between surface PM2.5 and AOD is not always linear. It is affected by multiple factors, such as the relative humidity since the AOD can be enhanced by aerosol swelling effects but the PM2.5 does not. Meanwhile, the correlation between AOD and surface level PM2.5 significantly depends on the

aerosol vertical distribution and aerosol particle size distribution. Thus, the uncertainties in those analysis and the influences on the results should be discussed.

The existing manuscript was not clear about the timescale when the word "constant" was used. What was meant is that the $PM_{2.5}$/AOD ratio is assumed to be constant from year to year (based on annually averaged ratios). This ratio can even change from year to year if there were an increasing trend in surface-level aerosol emissions. In the revised manuscript, we have devised a better way to compare trends: the POLDER/PARASOL and MODIS Deep Blue AOD offsets, determined from the AERONET validation at Fort McMurray, are corrected and then relative trends are used for $PM_{2.5}$ and satellite AOD. Thus, the $PM_{2.5}$/AOD ratio is not used in the revised manuscript. The Fort McMurray AERONET site is used for bias correction since it has temporal overlap with both sensors and has a longer record than the Fort McKay site. There is qualitative agreement on the magnitude of the offset at both sites for MODIS DB.

This is now described at p4L12:

For temporal trends, a simple linear regression is performed on relative anomalies derived from bias-corrected annual average and median AODs. The bias correction involves subtracting the AOD offset obtained through the validation with coincident Fort McMurray AERONET data.

P6, Line 28: Is this trend statistical significant?

Yes, the MODIS/Aqua DB and POLDER/PARASOL trends are both statistically significant. We will add "statistically" to the sentence as follows:

In fact, two satellite data products, namely POLDER/PARASOL and MODIS/Aqua DB, exhibit a statistically significant positive trend in this mining area.